# Mechanical properties of atomically thin boron nitride and the role of interlayer interactions

Aleksey Falin[1], Qiran Cai[1], Elton J.G. Santos[2,3], Declan Scullion[2], Dong Qian[4], Rui Zhang[4,5], Zhi Yang[6], Shaoming Huang[6], Kenji Watanabe[7], Takashi Taniguchi[7], Matthew R. Barnett[1], Ying Chen[1], Rodney S. Ruoff[8,9,10] & Lu Hua Li[1]

Atomically thin boron nitride (BN) nanosheets are important two-dimensional nanomaterials with many unique properties distinct from those of graphene, but investigation into their mechanical properties remains incomplete. Here we report that high-quality single-crystalline mono- and few-layer BN nanosheets are one of the strongest electrically insulating materials. More intriguingly, few-layer BN shows mechanical behaviours quite different from those of few-layer graphene under indentation. In striking contrast to graphene, whose strength decreases by more than 30% when the number of layers increases from 1 to 8, the mechanical strength of BN nanosheets is not sensitive to increasing thickness. We attribute this difference to the distinct interlayer interactions and hence sliding tendencies in these two materials under indentation. The significantly better interlayer integrity of BN nanosheets makes them a more attractive candidate than graphene for several applications, for example, as mechanical reinforcements.

[1] Institute for Frontier Materials, Deakin University, Geelong Waurn Ponds Campus, Waurn Ponds, Victoria 3216, Australia. [2] School of Mathematics and Physics, Queen's University Belfast, Belfast BT7 1NN, UK. [3] School of Chemistry and Chemical Engineering, Queen's University Belfast, Belfast BT9 5AL, UK. [4] Department of Mechanical Engineering, The University of Texas at Dallas, Richardson, Texas 75080, USA. [5] School of Astronautics, Northwestern Polytechnical University, Xi'an 710072, China. [6] Nanomaterials and Chemistry Key Laboratory, Wenzhou University, 276 Xueyuan Middle Road, Wenzhou, Zhejiang 325027, China. [7] National Institute for Materials Science, Namiki 1-1, Tsukuba, Ibaraki 305-0044, Japan. [8] Center for Multidimensional Carbon Materials, Institute for Basic Science (IBS), Ulsan 44919, Republic of Korea. [9] Department of Chemistry, Ulsan National Institute of Science and Technology (UNIST), Ulsan 44919, Republic of Korea. [10] School of Materials Science and Engineering, Ulsan National Institute of Science and Technology (UNIST), Ulsan 44919, Republic of Korea. Correspondence and requests for materials should be addressed to Y.C. (email: ian.chen@deakin.edu.au) or to L.H.L. (email: luhua.li@deakin.edu.au).

Two-dimensional (2D) nanomaterials, such as graphene, boron nitride (BN) and molybdenum disulfide (MoS₂) nanosheets, have many fascinating properties that could be useful for a wide range of applications, such as composite, nanoelectromechanical systems and sensing. Investigations on the mechanical properties of these nanomaterials are, therefore, essential. In this regard, the mechanical properties of monolayer (1L) graphene have been systematically studied. Although the reported experimental values of the elastic modulus of high-quality graphene vary between 0.5 and 2.4 TPa (refs 1–5), most studies obtained a value of $\sim 1$ TPa, that is, an effective Young's modulus ($E^{2D}$) of $\sim 342$ N m$^{-1}$ with an effective thickness of 0.335 nm, consistent with many theoretical calculations[6–8]. The theoretical and experimental fracture strengths of graphene are in the range of 70–130 GPa, and the intrinsic strain is between 14 and 33% (refs 1,7–9). It has been found that although low levels of defects do not have a negative influence on the elastic modulus of graphene[10,11], their presence can greatly deteriorate its strength[10,12–14]. The effect of grain boundaries in graphene has also been studied theoretically and experimentally[9,15–18]. As for the mechanical properties of few-layer graphene, it has been found that both the Young's modulus and strength of graphene decrease with increased thickness[19–24]. This has been explained by strong in-plane covalent bonding bonds and weak van der Waals interactions between the layers[22,25]. The mechanical properties of many other 2D nanomaterials, including MoS₂, tungsten disulfide (WS₂) and phosphorene, have also been studied[26–29].

BN nanosheets, which are composed of atomically thin hexagonal boron nitride (hBN), have a structure similar to graphene but possess many distinguished properties[30]. They, sometimes called white graphene, are insulators with bandgaps close to 6 eV. BN nanosheets can serve as dielectric substrates for graphene, MoS₂ and other 2D nanomaterials[31,32]. In addition, BN nanosheets are efficient emitters of deep ultraviolet light[33,34]. Moreover, monolayer BN is stable up to 800 °C in air[35]; in contrast, graphene starts to oxidize at 300 °C under the same conditions[36]. Therefore, BN nanosheets are candidates for reinforcing ceramic and metal matrix composites, which are normally fabricated at high temperatures. BN nanosheets can also be used in polymer composites when electrical insulation, optical transparency and enhanced thermal stability are desired. The thermal and chemical inertness of BN nanosheets are also ideal for corrosion protection at high temperatures[37,38]. Furthermore, BN nanosheets have a special surface adsorption capability[39] and can provide high sensitivity and reusability in sensing applications[40,41].

There have been a few measurements on few-layer BN produced by chemical vapour deposition (CVD), but the mechanical properties of monolayer BN have never been experimentally examined. Song et al. first reported that the elastic modulus of CVD-grown bilayer BN nanosheets was 0.334 ± 0.024 TPa (that is, $E^{2D} = 112 \pm 8$ N m$^{-1}$), and their fracture strength was 26.3 GPa (that is, 8.8 N m$^{-1}$)[42]. These values are much smaller than those predicted by theoretical calculations. From the aspect of theoretical calculations, although the mechanical properties of few-layer BN have never been theoretically investigated, the Young's modulus of 1L BN was predicted to be 0.716–0.977 TPa (that is, $E^{2D} = 239$–326 N m$^{-1}$ with an effective thickness of 0.334 nm), while its breaking strength fell in the wide range of 68–215 GPa (that is, 23–72 N m$^{-1}$)[42–51]. The degraded mechanical properties of the 2L CVD BN reported by Song et al. were attributed to the presence of defects and grain boundaries[52,53]. Kim et al. measured the Young's modulus of $\sim 15$ nm-thick (that is, $\sim 45$L) BN nanosheets produced by CVD to be 1.16 ± 0.1 TPa

(ref. 54). Li et al. investigated the bending modulus of $\sim 50$ nm-thick (that is, $\sim 150$L) BN nanosheets[55]. The lack of systematic study of the intrinsic mechanical properties of atomically thin BN of different thicknesses greatly hinders the study and use of these nanomaterials. On the other hand, the different interlayer interactions in few-layer BN and graphene[56–58] could play important roles in their mechanical properties.

Here, the mechanical properties of high-quality mono- and few-layer BN are experimentally revealed, to our knowledge, for the first time. The monolayer BN is found to have a Young's modulus of 0.865 ± 0.073 TPa, and fracture strength of 70.5 ± 5.5 GPa. In contrast to graphene, whose strength decreases dramatically with an increase in thickness, few-layer BN nanosheets (at least up to 9L) have a strength similar to that of 1L BN. Detailed theoretical and experimental investigations indicate that the difference is caused by the distinct interlayer interactions in these two nanomaterials under large in-plane strain and out-of-plane compression. This study suggests that BN nanosheets are one of the strongest insulating materials, and more importantly, the strong interlayer interaction in BN nanosheets, along with their thermal stability, make them ideal for mechanical reinforcement applications.

## Results

**Preparation and characterization of atomically thin BN.** The BN nanosheets were mechanically exfoliated from high-quality hBN single crystals[59] on 90 nm-thick silicon oxide covered silicon (SiO₂/Si) substrates with pre-fabricated micro-wells of 650 nm in radius. Figure 1a shows the optical microscopy image of a 1L BN covering seven micro-wells, and the corresponding atomic force microscopy (AFM) image is displayed in Fig. 1b. According to the height trace, the thickness of the 1L BN was 0.48 nm (Fig. 1c). The thickness of 2L and 3L BN was about 0.85 and 1.02 nm, respectively (see Supplementary Fig. 1). Figure 1d shows the Raman spectrum of the suspended part of the 1L BN, and its G band frequency centred at 1,366.5 cm$^{-1}$, which is very close to that of bulk hBN (that is, 1,366.4 cm$^{-1}$)[60]. For comparison purposes, mono- and few-layer graphene were also produced following the same method (see Supplementary Figs 2 and 3).

**Mechanical tests by indentation.** The mechanical properties of the mono- and few-layer graphene and BN nanosheets were studied by indentation at the centre of the suspended regions using AFM. To obtain load–displacement curves, the AFM displacements were converted into the deflection ($\delta$) of the nanosheets, as follows:

$$\delta = \Delta Z_{piezo} - \delta_{tip} \qquad (1)$$

where $\delta_{tip}$ is the deflection of the AFM tip; $\Delta Z_{piezo}$ is the z displacement of the AFM piezo/sample[4]. The deflection of 2D nanomaterials during indentation can be divided into two regions. Under a relatively small uniaxial load, the isotropic elastic response of 2D nanomaterials is linear; when the load and deformation are large, the load–displacement relation becomes cubic[1,19]. Therefore, the total load–displacement relationship in 2D nanomaterials during indentation includes both the linear and cubic terms[1]:

$$F = \sigma_0^{2D}(\pi a)\left(\frac{\delta}{a}\right) + E^{2D}(q^3 a)\left(\frac{\delta}{a}\right)^3 \qquad (2)$$

where $F$ is the applied load; $\sigma_0^{2D}$ is the 2D pre-tension of the nanosheet; $\delta$ is the deflection of the nanosheet under load $F$; $a$ is the radius of the micro-well; $q = 1/(1.049 - 0.15v - 0.16v^2)$ is a dimensionless constant; and $v$ is Poisson's ratio. $E^{2D}$ is the 2D

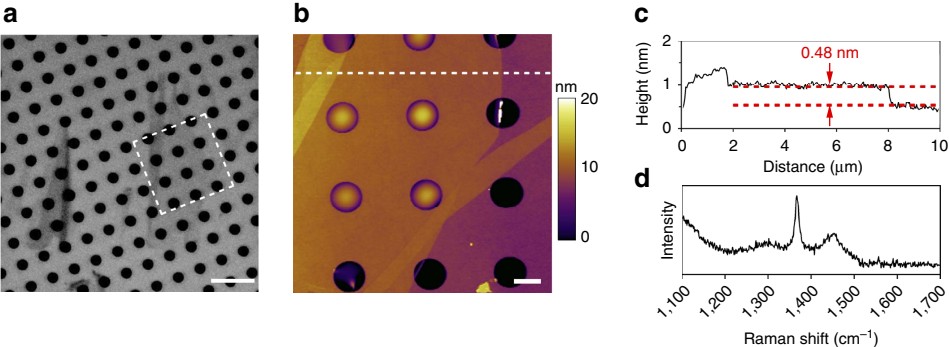

**Figure 1 | Characterization of 1L BN.** (**a**) Optical microscopy image of a 1L BN on a SiO₂/Si substrate with micro-wells of 1.3 μm in diameter; (**b**) AFM image of the BN nanosheet marked in the square of **a**; (**c**) the corresponding height trace of the dashed line in **b**; (**d**) Raman spectrum of the suspended part of the 1L BN. Scale bars 5 μm in **a** and 1 μm in **b**.

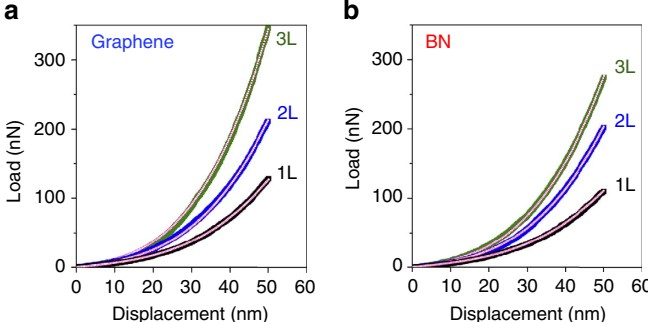

**Figure 2 | Load-displacement curves and the corresponding fittings.** (**a**) 1-3L graphene and (**b**) 1-3L BN nanosheets.

effective Young's modulus of the nanosheet, which can be converted to the conventional bulk (that is, volumetric) modulus ($E$) by dividing it by the thickness of the nanosheet. For BN, we used an effective thickness of 0.334 nm, and a Poisson ratio of 0.211 (refs 43,45); for graphene, the effective thickness was 0.335 nm, and the Poisson ratio was 0.165 (ref. 1). The elastic moduli of atomically thin BN and graphene could be deduced by fitting the loading curves using equation 2 (refs 1,26,42). Typical loading curves of 1–3L graphene and BN nanosheets till a displacement of ~50 nm, and the corresponding fittings, are compared in Fig. 2.

**Elastic modulus and breaking strength.** Figure 3 summarizes the Young's moduli of graphene and BN nanosheets of different thicknesses. The $E^{2D}$ of 1–3L graphene were $342 \pm 8$ N m$^{-1}$ ($N=11$), $645 \pm 16$ N m$^{-1}$ ($N=13$) and $985 \pm 10$ N m$^{-1}$ ($N=6$), respectively. These values are consistent with those obtained by previous studies using AFM[1,19]. The average $E^{2D}$ of 1L BN was $289 \pm 24$ N m$^{-1}$ ($N=11$). This result is in agreement with a few theoretical predictions[43–46,51]. The $E^{2D}$ of 2L and 3L BN nanosheets were $590 \pm 38$ N m$^{-1}$ ($N=14$) and $822 \pm 44$ N m$^{-1}$ ($N=6$), respectively. The dashed lines in Fig. 3a show the projections of the $E^{2D}$ of graphene and BN nanosheets with increased thickness, which were obtained by multiplying the $E^{2D}$ values of their monolayers by the number of layers. In other words, the difference in the experimental data and dashed lines indicates the relative changes of $E^{2D}$ with the increased thickness of graphene and BN. It can be seen that the $E^{2D}$ of graphene deviated more than that of BN as thickness increased. This can be shown more clearly by plotting the (volumetric) Young's moduli of graphene and BN at different thicknesses (Fig. 3b). The $E$ of 1L

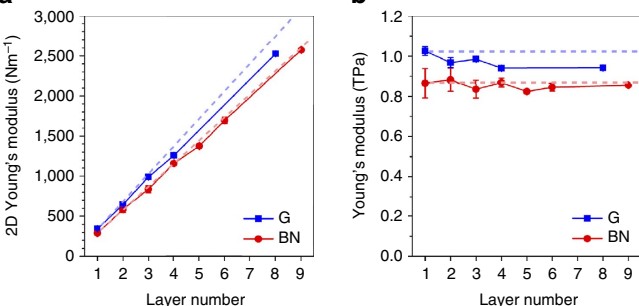

**Figure 3 | Elastic properties of graphene and BN nanosheets.** (**a**) 2D Young's modulus ($E^{2D}$) of graphene (G) and BN nanosheets of different thicknesses, along with the dashed projections calculated based on multiplying the number of layers by the $E^{2D}$ of the monolayers; (**b**) Volumetric Young's modulus ($E$) of graphene and BN nanosheets of different thicknesses, along with dashed lines that show the Young's moduli of 1L graphene and BN.

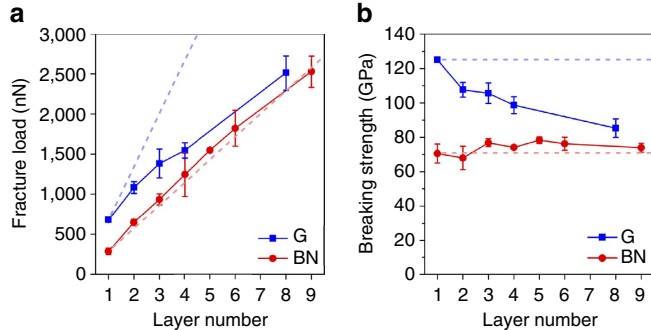

**Figure 4 | Mechanical strengths of graphene and BN nanosheets.** (**a**) Fracture load and (**b**) breaking strength of graphene and BN of different thicknesses. The dashed lines in **a** are the projections of the fracture load of BN and graphene (G) of different thicknesses based on the multiplication of the strength of their monolayers by the number of layers.

graphene was $1.026 \pm 0.022$ TPa, but that of 8L graphene was reduced to $0.942 \pm 0.003$ TPa. The $E$ values of 1L and 9L BN nanosheets were quite similar: $0.865 \pm 0.073$ and $0.856 \pm 0.003$ TPa, respectively.

The strengths of graphene and BN nanosheets of different thicknesses were calculated based on load–displacement curves and fracture loads using finite element simulation. The fracture

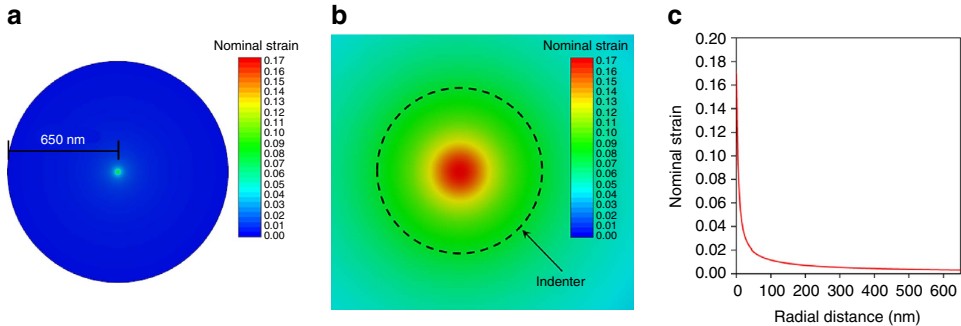

**Figure 5 | Strain distribution obtained from finite element simulations.** (**a**) Nominal strain distribution in a 1L BN suspended over a micro-well with a radius of 650 nm under a fracture load; (**b**) enlarged view close to the indentation centre, with the indenter tip (6.3 nm in radius) shown as a dashed circle; (**c**) strain distribution along the radius of the nanosheet under the fracture load.

loads ($F_f$) of graphene and BN of different thicknesses are shown in Fig. 4a. Similar to Fig. 3a, the dashed lines in Fig. 4a are the projections calculated by multiplying the fracture load of 1L graphene and BN by the number of layers. The fracture loads of multilayer graphene deviated more from the blue dashed line as the thickness increased. For example, the fracture load of 8L graphene was 53.7% smaller than eight times the fracture load of the 1L graphene. In contrast, the fracture loads of BN of different thicknesses closely followed the red dashed line. These different trends are also shown in their mechanical strengths. As shown in Fig. 4b, the breaking strengths of graphene were $125.0 \pm 0$ GPa (that is, 2D strength of $41.9 \pm 0$ N m$^{-1}$), $107.7 \pm 4.3$ GPa ($72.1 \pm 2.9$ N m$^{-1}$), $105.6 \pm 6.0$ GPa ($106.2 \pm 6.0$ N m$^{-1}$), and $85.3 \pm 5.4$ GPa ($228.6 \pm 14.5$ N m$^{-1}$) for monolayer, bilayer, trilayer, and eight layers, respectively. Again, these values are in agreement with those reported previously[19,22]. According to these values, no defect was present in the part of graphene close to the indentation center[9,10]. The strengths of 1-3L BN were $70.5 \pm 5.5$ GPa ($23.6 \pm 1.8$ N m$^{-1}$), $68.0 \pm 6.8$ GPa ($45.4 \pm 4.5$ N m$^{-1}$), and $76.9 \pm 2.3$ GPa ($77.0 \pm 2.3$ N m$^{-1}$), respectively. Previous theoretical calculations yielded a quite different breaking strength for 1L BN, and our experimental results match well the value calculated by Peng *et al.* using density functional theory (DFT)[45], but are much smaller than those predicted by Han *et al.* and Mortazavi *et al.*, both of which used molecular dynamics simulations[43,49].

The finite element simulations were also used to resolve the strain distribution in BN under a fracture load. Figure 5 shows the nominal strain distribution in a 1L BN. The maximum strain occurred at the very centre of the load. That is, only a small portion of the BN under and adjacent to the indenter tip (dashed circle in Fig. 5b) was highly strained, and the behaviour of the rest of the nanosheet was almost linear elastic. This can be also seen from the strain distribution curve along the 650 nm radius of the suspended nanosheet (Fig. 5c). The maximum nominal strain in this 1L BN was $\sim 17\%$. Similarly to the trend of the strength, the averaged maximum nominal strain in BN of different thicknesses was quite close: $12.5 \pm 3.0\%$ for 1L BN and $13.3 \pm 1.7$ for 9L BN.

**Changed sliding energy under indentation.** Our results show that the strength of graphene largely decreased as the thickness increased. According to previous reports, this is due to interlayer slippage in few-layer graphene during indentation[22]. A similar phenomenon has also been observed from $MoS_2$/graphene and $MoS_2$/$WS_2$ heterostructures: the 2D Young's modulus and strength of heterostructures were smaller than the sum of those from each component[28]. However, the strength of the BN nanosheets remained constant over different thicknesses (Fig. 4).

This difference between graphene and BN could be caused by different interlayer interactions in these nanomaterials despite of their analogous structure. We used *ab initio* DFT calculations, including van der Waals interactions, to study the sliding energy in bilayer graphene and BN. According to the finite element simulations (Fig. 5), most of the suspended nanosheets (not close to the indentation centre) experienced a very small in-plane strain and no out-of-plane compression even under the fracture load. The sliding energy in standard or equilibrium 2L graphene and BN can thus represent the interlayer interaction in the low-strained parts of the nanosheets. However, the small portions of graphene and BN nanosheets close to the indentation centre were under a large in-plane tensile strain and out-of-plane compression, and there has been no study on how strain and compression affect their interlayer sliding. Figure 6a shows the finite element calculated strain distribution (solid lines) and out-of-plane pressure (dashed lines) in 2L graphene and BN within a radial distance of 10 nm from the indentation centre under their fracture loads. In the vdW-corrected DFT calculations, we chose four combinations of bi-axial strain and hydraulic pressure conditions to reveal the interlayer interactions close to the indentation centre of 2L graphene and BN. The sliding energy was taken from the total energy differences relative to AB to AB or AA′ to AA′ positions at different points of the sliding pathway[56,57]. The four conditions correspond to radial distances of 0, 2, 4 and 10 nm away from the indentation centre (grey vertical dotted lines in Fig. 6a), and the strain plus pressure values are hence 21.7% + 16.9 GPa (at a radial distance of 0 nm or the indentation centre), 16.8% + 17.9 GPa (2 nm), 12.4% + 8.3 GPa (4 nm), and 7.2% + 0 GPa (10 nm) for 2L graphene; and 14.5% + 14.1 GPa (0 nm), 12.4% + 14.2 GPa (2 nm), 9.8% + 5.3 GPa (4 nm), and 5.7% + 0 GPa (10 nm) for 2L BN, respectively.

In standard or equilibrium crystal lattices (that is, without strain or compression), the sliding energy from the AA′ to AA′ stacking in 2L BN was only slightly larger than that from the pre-defined zero sliding energy of the AB to AB stacking in 2L graphene, that is, 7.22 versus 5.12 meV per unit cell (Fig. 6b). These results are consistent with previous calculations[56,57], even though numerical differences were observed because a different methodology was adopted in the description of the local chemical environment of the atoms. When 2L graphene and BN were strained without out-of-plane pressure (that is, at a radial distance of 10 nm from indentation centre, as shown in Fig. 6a), both of their sliding energies increased: 11.64 meV for graphene, and 21.57 meV for BN (Fig. 6c). Under further increased strain and pressure, 2L graphene and BN started to show a very different sliding tendency. At a radial distance of 4 nm, the sliding energy in 2L graphene reduced to almost zero, that is, 0.92 meV per unit

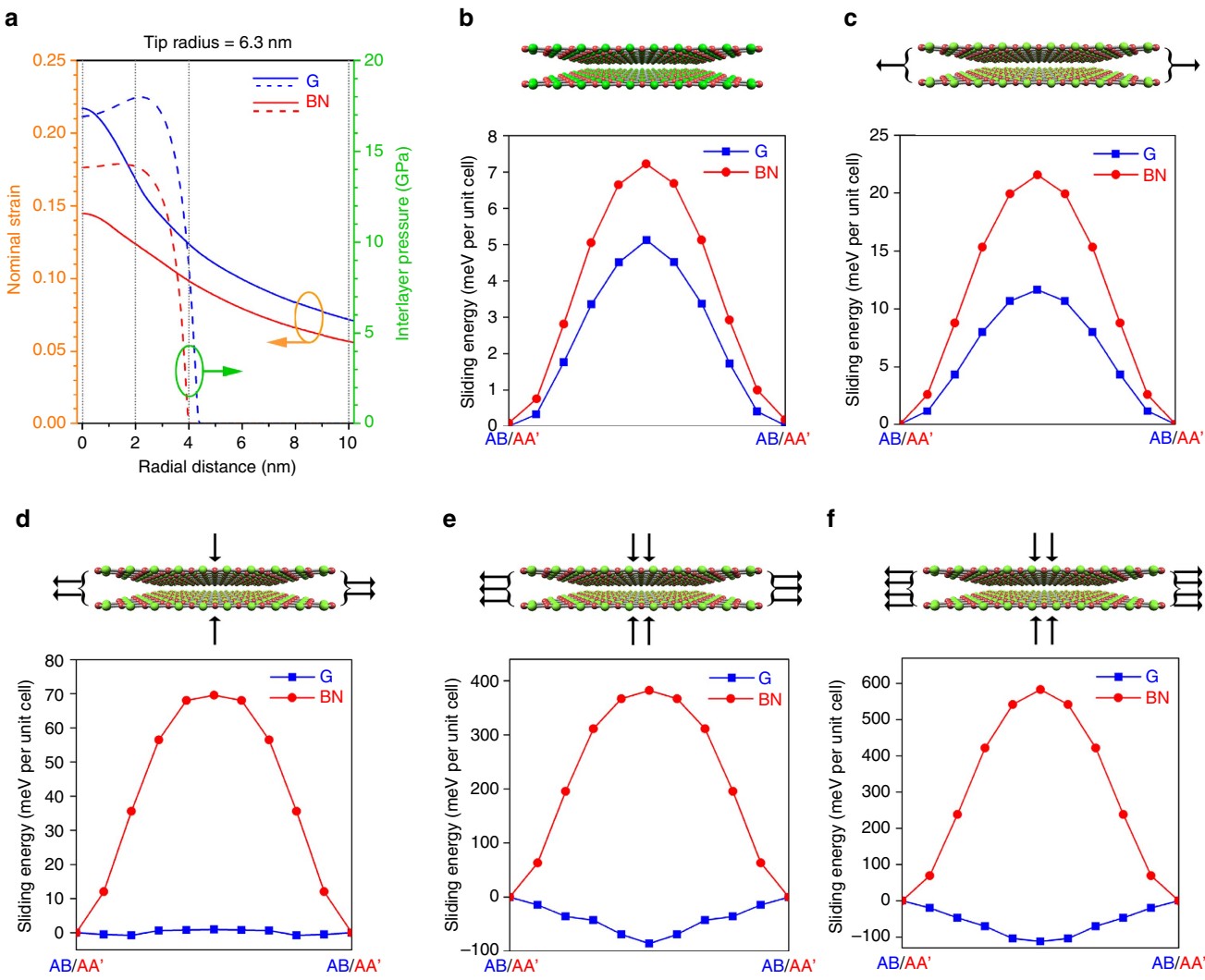

**Figure 6 | Changed sliding energies in 2L graphene and BN due to strain and pressure.** (**a**) Finite element calculation induced in-plane strain (solid lines) and out-of-plane pressure (dashed lines) in 2L graphene and BN within a radial distance of 10 nm from the indentation centre; (**b-f**) sliding energies in 2L graphene (AB to AB) and BN (AA′ to AA′) under five conditions: (**b**) equilibrium/standard state without strain or pressure, representing the portion of graphene and BN not close to the indentation centre; (**c**) at a radial distance of 10 nm away from indentation centre: 7.2% strain + 0 GPa pressure in 2L graphene, and 5.7% strain + 0 GPa pressure in 2L BN; (**d**) 4 nm away from indentation centre: 12.4% strain + 8.3 GPa pressure in 2L graphene and 9.8% strain + 5.3 GPa pressure in 2L BN; (**e**) 2 nm away from indentation centre: 16.8% strain + 17.9 GPa pressure in 2L graphene and 12.4% strain + 14.2 GPa pressure in 2L BN; (**f**) the indentation centre or 0 nm: 21.7% strain + 16.9 GPa pressure in 2L graphene and 14.5% strain + 14.1 GPa pressure in 2L BN.

cell; while that in 2L BN further increased to 69.56 meV per unit cell (Fig. 6d). Within a radial distance of 0–2 nm, the difference became more prominent: the sliding energy in graphene was as small as −112.26 meV per unit cell, but that in BN was as large as 582.84 meV per unit cell (Fig. 6e,f). To validate the above results, we also performed simulations at a higher level of theory using DFT (PBE) plus many-body dispersion (MBD) corrections (PBE + MBD)[61–64]. The PBE + MBD results which are shown in Supplementary Fig. 8 are fully consistent with those from optB88-vdW functional and previous works under the zero strain and pressure condition, though numerical differences as high as ∼30% in sliding energies between optB88-vdW and PBE + MBD approaches were observed, which evince the accuracy of our simulations. This comparison suggests the generality of the underlying physics associated with the sliding processes, which are not method- or functional-dependent. In addition, we found that there was an interesting interplay between strain and pressure in affecting the sliding energy in graphene and BN (see Supplementary Figs 9 and 10). The rather different electronic

characters of graphene and BN, that is, semi-metallic and insulating, respectively, play a major role in their distinct sliding energies. When large strain and pressure are applied on graphene, its $2p_z$ orbitals tend to overlap; the more polar character of those orbitals in BN, on the other hand, localizes the electronic density (to be published). This difference results in opposite changes in sliding energy in the two materials under strain and pressure. These results indicate that the BN layers close to the indentation centre were strongly glued and very unlikely to develop interlayer sliding. In striking contrast, the graphene layers could spontaneously slide between each other as the AB stacking was no longer stable.

**Different sliding tendencies in BN and graphene.** For simplicity, hypothetical sandwich beam geometries were used to explore the sliding tendencies in 2L graphene and BN. Note that such an estimation did not consider the nonlinear deformation in the structures under indentation. The two surface layers of the 2L

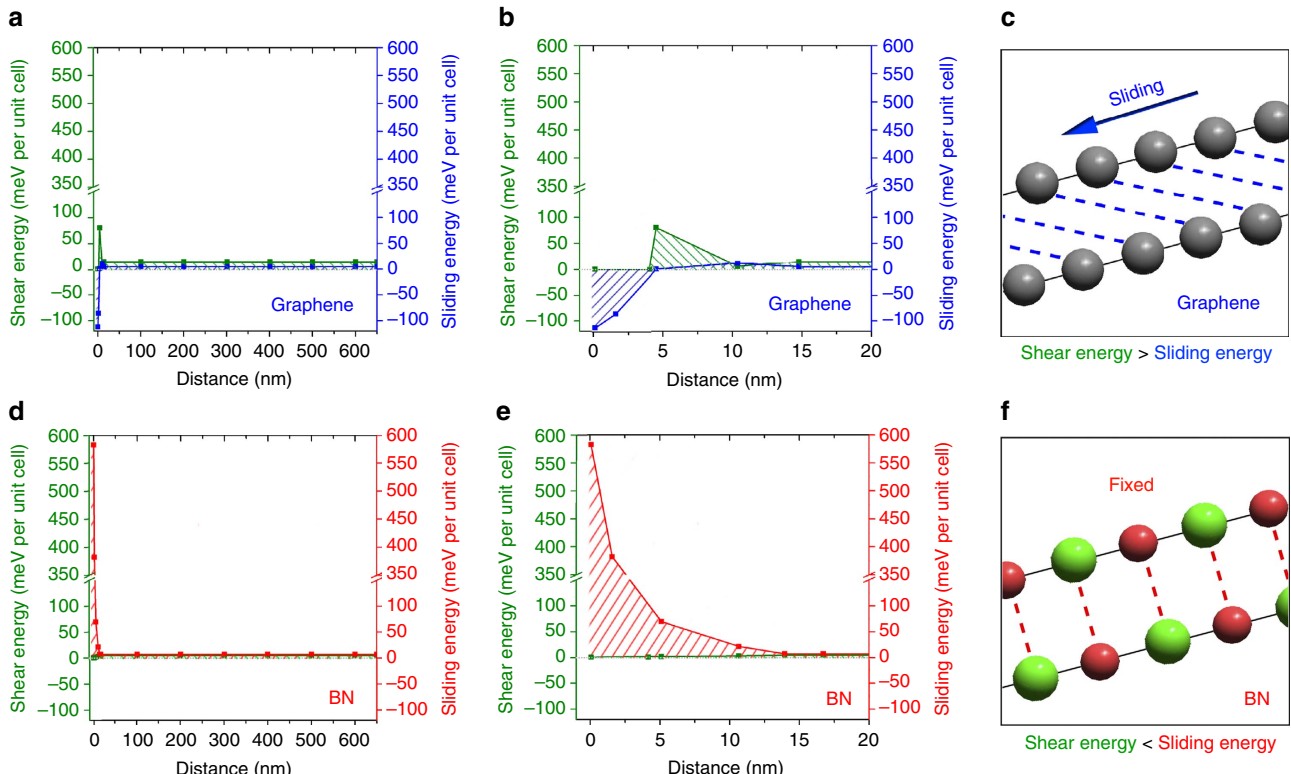

**Figure 7 | Distributions of the shear strain energy and sliding energy barrier in graphene and BN under fracture loads.** (**a**) Comparison between the shear strain energy in the sandwich beam structure and sliding energy calculated by vdW-DFT simulations in 2L graphene; (**b**) the enlarged view of the region close to the indentation centre; (**c**) diagram showing that slippage happens easily in graphene; (**d**) comparison between the shear energy in the sandwich beam structure and sliding energy calculated by vdW-DFT simulations in 2L BN; (**e**) the enlarged view of the region close to the indentation centre; (**f**) diagram showing that interlayer sliding is unlikely to occur in the case of BN.

graphene and BN can be defined as faces, and the interlayer interactions including van der Waals interactions can be viewed as a core. Such designation meets the basic requirement for a sandwich structure where the faces are much stiffer than the core. In addition, the core in graphene and BN nanosheets satisfies the concept of an 'antiplane' core, which has no contribution to the bending stiffness of the structure but can sustain a finite shear stress. The beams with a length of 1,300 nm and width of the unit cell of graphite and hBN have both ends clamped and are under central loads ($F'$). In the isotropic elastic limit, the shear strain energy in the core ($U_{shear}$) of the sandwich beam structures can be given as:

$$U_{shear} = \frac{AG}{2} \int \gamma^2 \, dx \qquad (3)$$

where $\gamma$ is shear strain, which can be calculated by $\gamma = d/c \cdot dw_s/dx$ ($d$ is the separation of the faces, and $c$ is the thickness of the core); $x$ is the distance to the central point of load ($F'$); $w_s$ is shear displacement, which is equal to $F'x/4AG$. AG is the shear stiffness of the sandwich structure. The shear stiffness of graphene and BN was linearly approximated based on the vdW-DFT-deduced sliding energy from the AB to AB stacking in graphene, and from the AA′ to AA′ stacking in BN. When no strain or compression was applied, the $G$ values of graphene and BN were 5.11 and 6.61 GPa, respectively (see Supplementary Information, Note 3). These values are in the range of previously reported values: 0.7 − 15.4 GPa for graphene/graphite[65–69], and 2.5 − 9 GPa for hBN[44,70–72]. However, under a large strain and compression close to the indentation centre, the $G$ value of graphene became zero or even negative, and that of BN increased enormously to 534 GPa. It should be noted that we deem that the

shear strain energy became zero directly under the loads. Therefore, the shear strain energy distributed over the distance of the sandwich beam structures of graphene and BN could be estimated (see Supplementary Information, Note 2).

Figure 7a,d compares the distribution of the shear strain energy (from the sandwich beam theory) and sliding energy (from the vdW-DFT simulations) in the 2L graphene and BN beams. The overall sliding energy in graphene over the 650 nm semi-length distance (that is, the shaded area in blue in Fig. 7a) was smaller than that in BN (that is, the shaded area in red in Fig. 7d), but the shear strain energy in graphene was much larger than that in BN (that is, the shaded areas in green), especially close to the indentation centre. These differences were partly due to the different sliding energies in the two materials, which, in turn, affected the local shear modulus and shear strain energy. It can be seen that under the conditions considered, BN displays a sliding energy larger than the shear strain energy over the beam. Figure 7b,e shows the enlarged views of the regions close to the indentation centre. The results illustrate the tendency for 2L graphene to experience sliding but for 2L BN to resist sliding. The quantitative sliding encountered in our experiments must wait for future analysis, but the different tendencies between the graphene and BN for sliding were important. Sliding could concentrate stresses in the lowest layer bonded strongly to the SiO$_2$ substrate[22]. This effectively shielded the other layers. As a result, any extra layers added to graphene did not proportionally add to the load bearing capacity. Such stress concentration and concomitant shielding were less likely to be present in BN. Thus, the addition of layers to BN tended to make a proportional addition to the load bearing capacity, giving rise to fracture loads linearly increased with the numbers of layers.

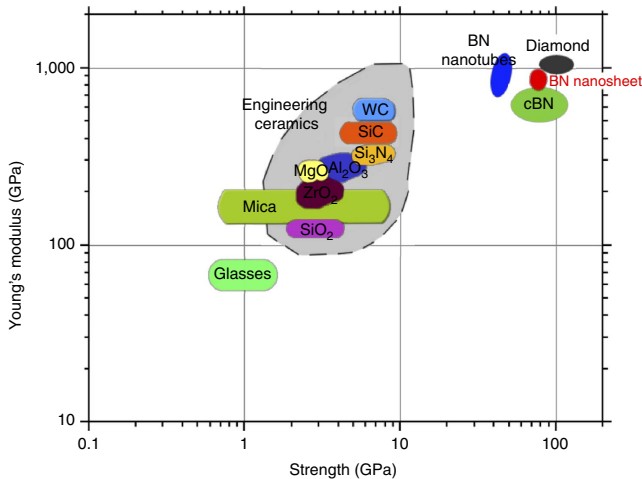

**Figure 8 | Modulus-strength graph.** The mechanical properties of different electrically insulating materials, including monolayer and few-layer BN, are compared.

## Discussion

We experimentally measured the mechanical properties of high-quality 1–9L BN nanosheets using AFM. Monolayer BN had a Young's modulus of $0.865 \pm 0.073$ TPa, and a fracture strength of $70.5 \pm 5.5$ GPa. Few-layer BN was as strong as 1L BN. This was very different to the case of graphene whose modulus and strength were found to decrease dramatically with increased thickness. Our DFT calculations including van der Waals interactions revealed that 2L graphene had energetically favoured sliding under an in-plane strain and large compression close to the indentation centre, while 2L BN could have large positive sliding energies under the same conditions to prevent it from sliding. According to the simplified models using the sandwich beam structures, graphene layers tended to slide during indentation, but BN layers were mostly glued, especially the area under the tip. Thus, the different trends in modulus and strength between graphene and BN nanosheets with increasing thickness were caused by their dramatically different interlayer interactions. Our results show that BN nanosheets are one of the strongest electrically insulating materials (Fig. 8).

## Methods

**Materials and fabrication.** Mechanical exfoliation by Scotch tape was used to prepare suspended graphene and BN nanosheets[32,35]. For comparison purposes, the indentation and fitting procedures for graphene and BN nanosheets were identical. A Cypher AFM was employed for the indentation tests. Two cantilevers with diamond tips were used because of the high strength of the membranes. The spring constants of the cantilevers were determined using a combination of the thermal noise method and the Sader method. The tip's radii were 5.6 and 6.3 nm, measured by transmission electron microscope. The indentation processes were conducted in ambient conditions and performed on relatively large nanosheets to prevent inaccuracy caused by their slippage on the substrate. The loading and unloading velocity for all measurements was constant ($0.5 \, \mu m \, s^{-1}$). Loading/unloading curves with no obvious hysteresis were used for fitting and calculations; curves showing large hystereses were excluded. The loading/unloading curves of few-layer graphene could not be reproduced by equation (1), and hence were fitted till $\sim 50$ nm of deflection for calculating the Young's moduli.

**Finite element analysis.** Computational simulations were performed using the commercial nonlinear finite element code ABAQUS. The diamond tips were modelled as rigid spheres. The nanosheets were modelled as axisymmetric membranes with a radius of 650 nm. The initial thicknesses for graphene and BN nanosheets were assigned as $0.335 \times N$ nm and $0.334 \times N$ nm, respectively, where $N$ is the number of layers. A total of 1,663 two-node linear axisymmetric membrane elements (MAX1) were employed with mesh densities varying linearly from 0.1 nm (centre) to 1.0 nm (outermost). The interactions between the indenter tip and nanosheet were modelled by a frictionless contact algorithm. An indentation depth

of 100 nm was applied to a prescribed displacement of 0.1 nm per load step. The constitutive behaviours of both graphene and BN were assumed to be nonlinear elastic, and thus expressed as:

$$\sigma = E\varepsilon + D\varepsilon^2 \qquad (4)$$

where $E$ is Young's modulus and $D$ is the third-order elastic constant. The Young's moduli of graphene and BN were set to 1,000 GPa and 865 GPa, respectively. The value of $D$ for graphene was $-2,000$ GPa (ref. 1). The value of $D$ for BN was $-2,035$ GPa, which was obtained from experimental results. The nonlinear elastic constitutive behaviour was implemented in ABAQUS using an equivalent elastic-plastic material model as previously described[1]. To verify the nonlinear elastic effects, simulations using a linear elastic model were also performed by dropping the nonlinear term in equation (4) in the constitutive model. To compute the fracture strength, the load–displacement curves obtained from the finite element methods were compared with the corresponding experimental data, and the simulation loading steps corresponding to the point at which fracture took place were then identified based on the fracture loads from experiments. Subsequently, the fracture strength was obtained as a volume average of the stress values of the elements that were directly underneath the indenter at the corresponding loading steps. The interlayer pressure was approximated by the contact pressure between the indenter and nanosheets following the surface-to-surface contact model[73].

**van der Waals ab initio calculations.** The calculations reported here are based on the ab initio DFT using the VASP code[74,75]. The generalized gradient approximation[76] along with the optB88-vdW (ref. 77) functional was used, with a well-converged plane-wave cutoff of 1,100 eV. Calculations taken into account MBD corrections (PBE + MBD)[61–64] have also been performed to check any limitations of the optB88-vdW functional. Similar results were found using both methods. Projected augmented wave method[78,79] has been used in the description of the bonding environment for B, N and C. The atomic coordinates were allowed to relax until the forces on the ions were less than $1 \times 10^{-8}$ eV Å$^{-1}$ under the conjugate gradient algorithm. The electronic convergence was set to $1 \times 10^{-8}$ eV. The lattice constants for the monolayer BN unit cell were optimized and found to be a = 2.510 Å. To avoid any interactions between the supercells in the non-periodic direction, a 20 Å vacuum space was used in all calculations. The Brillouin zone was sampled with a $24 \times 24 \times 1$ grid under the Monkhorst-Pack scheme[80] to perform relaxations with and without van der Waals interactions. In addition to this, we used a Fermi-Dirac distribution with an electronic temperature of $k_B T = 20$ meV to resolve the electronic structure. Bi-axial strain and hydraulic pressure were used in the calculations.

**Data availability.** The data that support the findings of this study are available from the corresponding author on request.

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

## Acknowledgements

L.H.L. thanks the financial support from Australian Research Council (ARC) via Discovery Early Career Researcher Award (DE160100796). A.F. thanks Discovery Project (DP150102346) for scholarship. NSFC No. 51420105002 and No. 51572197, and R.S.R. by the Institute for Basic Science (IBS-R019-D1) are also acknowledged. E.J.G.S. acknowledges the use of computational resources from the UK national high-performance computing service, ARCHER, for which access was obtained via the UKCP consortium and funded by EPSRC grant ref EP/K013564/1; and the Extreme Science and Engineering Discovery Environment (XSEDE), supported by NSF grants TG-DMR120049 and TG-DMR150017. The Queens Fellow Award through the start-up grant number M8407MPH and the Sustainable Energy PRP are also acknowledged. D.S. thanks his EPSRC studentship. K.W. and T.T. acknowledge support from the Elemental Strategy Initiative conducted by the MEXT, Japan and JSPS KAKENHI Grant Numbers JP26248061, JP15K21722 and JP25106006. Dr Abu Sadek assisted the fabrication of the percolated Si wafer at the Melbourne Centre for Nanofabrication (MCN) in the Victorian Node of the Australian National Fabrication Facility (ANFF).

## Author contributions

L.H.L. conceived and directed the project. L.H.L., Q.C. and A.F. prepared the samples. A.F. performed indentation and analysed the data. E.J.G.S. and D.S. performed DFT calculations. D.Q. and R.Z. performed finite element simulations. K.W. and T.T. provided hBN single crystals. M.R.B., A.F. and L.H.L. estimated the shear strain energy. R.S.R., Z.Y. and S.H. discussed the results. Y.C. provided intellectual guidance and managed project finances. Y.C. and M.R.B. reviewed the progress periodically. L.H.L. and A.F. co-wrote the manuscript with input from all authors.

## Additional information

**Competing interests:** The authors declare no competing financial interests.

