## [Peer Review File · Nature Communications]

Reviewers' Comments:

Reviewer #1 (Remarks to the Author):

The authors measured the mechanical strength and Young's modules of BN of different layer numbers through AFM indentation. It is found that the volumetric Young's modules and breaking strength of BN are independent of the layer number, in contrast to that of graphene. The difference is proven to be attributed to the change of sliding energy under in-plane strain and out-of-plane compression. In general, the work is interesting and comprehensive. However, before the consideration of publication, several concerns should be clarified.

1. The reported results are not "intrinsic" difference between graphene and BN. They can only be observed in measurements where bending of the 2D materials is presence, such as the AFM indentation adopted here, and are not expected for pure-stretch measurements. This point should be keep in mind during re-organize the manuscript.
2. Experimental data and corresponding discussion of bulk materials (thickness >10 nm) are desired.
3. How many times the test was conducted for each layer number? And how about the variation?
4. Which is the factor that dominates the change of the sliding energy in graphene and BN? in-plane strain or out-of-plane compression? DFT simulations of pure compression should be made to reveal the mechanisms.

Reviewer #2 (Remarks to the Author):

This manuscript presents an experimental study of mechanical properties of h-BN layers, both in monolayer and multilayer regime. The key finding is that the mechanical response of h-BN layers is qualitatively different from multilayered graphene, despite the very similar atomistic structure of these two quasi-2D materials. This is attributed to the fact that h-BN layers "stick to each other" much stronger under applied load, while graphene seems to be prone to interlayer sliding, leading to a much less robust mechanical response.

The very different mechanical properties of h-BN and graphene

under applied load constitute a very interesting finding that will surely lead to substantial future work. I was certainly entertained while reading the manuscript and felt that I learnt a few new things. The manuscript is written very clearly and is accessible to a wide audience of chemists and materials scientists. Therefore, I recommend publication in Nature Communications.

I do have a few recommendations regarding the interpretation of DFT calculations that the authors should address before publication:

(1) The experiments seem to be very clear on the different mechanical response of h-BN and graphene. However, the DFT calculations invariably rely on several (many!) approximations, both in the modeling of the actual physical process and in the solution of the electronic structure equations. The authors compare to existing calculations in the literature, and their own calculations agree with some previous results, while they disagree with others. No analysis is attempted to explain this agreement/disagreement. I would have liked to see at least a certain degree of analysis to clarify the situation in the field.

It is often the case that results of "first principles" calculations are taken without any critical analysis. I would expect better from a paper in Nature Communications. The approximations employed in different methods are well known to experts and the authors should strive to make an effort in assessing the reliability of their own and related previous calculations.

(2) Connected to the previous point, the particular DFT functional that the authors employ (opt-B88-vdW) significantly overestimates interlayer interactions in graphene (and most likely in h-BN). This has been documented in <http://journals.aps.org/prl/abstract/10.1103/PhysRevLett.115.115501>, for further analysis of the complexities involved in modeling interlayer sliding in graphene and h-BN, see Ref. 69. Either the authors should explicitly mention that their approach is inherently very approximate, or it would even be better if they added calculations using a less approximate method, such as PBE+MBD to reproduce some of their interlayer sliding potentials. This method is available in the code employed by the authors. If it turns out that the findings remain the same, it would make the paper significantly more interesting for theoretical materials scientists.

(3) The authors comment that no previous study looked at mechanical

properties of stacked h-BN and their difference from graphene. However, this is precisely what Ref. 69 did for multilayered h-BN and graphene, albeit in absence of any applied load. Hence, this should be mentioned in the introduction to the paper.

Reviewer #3 (Remarks to the Author):

The manuscript describes the differences in mechanical properties and interlayer interaction difference under mechanical load. It shows that although graphene has thickness dependence, h-BN does not. The authors carefully investigated the mechanism behind this phenomenon using experimental and theoretical approaches and concluded that h-BN has stronger interlayer interaction under normal load and tensile strain than graphene. I find that the study is very systematic and clear and highly recommend to be published after a minor revision. I have a few comments for clarification of their arguments.

1. In Figure 1(c), the height profile is drawn in the opposite direction and is confusing to the readers. The profile direction should be reversed.
2. In line 220, they used DFT calculation for sliding energy. They must have applied strain on graphene and h-BN in the calculations. However, there is no mention on the strain directionality. Is it mono-axial or bi-axial strain calculation? Depending on the directionality, the calculation result will be very different.
3. They calculated sliding energy. But there is no mention on the definition of “sliding energy”. They should define it and meaning in this study.

Dear Editor,

We thank the valuable comments from the Reviewers and have made proper changes in the revised version accordingly. Please see below for the details.

Reviewer #1 (Remarks to the Author):

The authors measured the mechanical strength and Young's modules of BN of different layer numbers through AFM indentation. It is found that the volumetric Young's modules and breaking strength of BN are independent of the layer number, in contrast to that of graphene. The difference is proven to be attributed to the change of sliding energy under in-plane strain and out-of-plane compression. In general, the work is interesting and comprehensive. However, before the consideration of publication, several concerns should be clarified.

1. The reported results are not "intrinsic" difference between graphene and BN. They can only be observed in measurements where bending of the 2D materials is presence, such as the AFM indentation adopted here, and are not expected for pure-stretch measurements. This point should be keep in mind during re-organize the manuscript.

Response and changes: We thank the reviewer for the comments. The word “intrinsic” is to describe the mechanical properties of atomically thin BN *without the effects of defects*, but we agree with the reviewer that readers may interpret it in a different way. To avoid possible confusion, we have removed the word “intrinsic” in the manuscript.

i) P.1 in Abstract: “intrinsic” has been deleted: “However, the lack of the knowledge of their mechanical properties greatly hindered their applications.”

ii) P.3 the last sentence of the first paragraph: “intrinsic” has been deleted: “The mechanical properties of many other 2D nanomaterials, including MoS₂, tungsten disulfide (WS₂), and phosphorene have also been studied.”

iii) P.5 the first sentence of the second paragraph: “intrinsic” has been deleted and “high-quality” has been added: “Here, the mechanical properties of high-quality mono- and few-layer BN are experimentally revealed for the first time.”

2. Experimental data and corresponding discussion of bulk materials (thickness >10 nm) are desired.

Response: Following the reviewer’s suggestion, we have carried out new mechanical measurements on ~10nm-thick graphene (30L) and BN (32L) and estimated the corresponding Young’s modulus using the same membrane model (see Figure R1). The main issue with these values is that the membrane model is no longer a good approximation at such thicknesses due to the significant effects of bending, especially that the thickness is comparable to the radius of the indenter tip now. To illustrate this, we performed additional FEM simulations to study the effects of thickness, as shown in Figure R2. The simulation reveals that when a sheet is thin (e.g. 1L), the stresses in the imaginary “top” and “bottom” layers are very close so that the membrane

model works very well. In contrast, when the thickness of a sheet increases to $32L$, the stress on the top and bottom surfaces becomes dramatically different, i.e. stretch in top (positive stress) and compression in bottom (negative stress) (Figure R2 right). Because of this, no good fitting can be achieved using the membrane model when the fitting was applied to the loading curves up to fracture (Figure R3a); therefore, the Young's moduli of the $\sim 10\text{nm}$ sheets shown in Figure R1 were deduced from the fittings up to much smaller loads/displacements (Figure R3b). Due to the fact that the stress conditions are dramatically different between the thin and thick layers (i.e. the ignorance of the bending effect), it is misleading to directly compare the Young's modulus of $1-9L$ sheets with that of the $\sim 30L$ sheets, calculated both using the membrane model. Since the focus of this manuscript is the mechanical properties of atomically thin BN, we do not include these new results to the manuscript.

In terms of strength of $\sim 10\text{nm}$ sheets, we were not able to get any strength values from FEM simulations, because, as discussed above, the membrane model in FEM which perfectly suits atomically thin sheets simply did not work for $\sim 10\text{nm}$ ones. That is, no convergence could be obtained in the FEM simulation using the membrane model, because it neglects the important bending effects in $30L$ and $32L$ thick sheets. As such, it clearly shows that the current model used for atomically thin sheets cannot be simply extended to the bulk crystals.

Figure R1. The comparison of the Young's modulus of 30L graphene and 32L BN with their mono- and few-layer sheets. All values were calculated using the membrane model.

Figure R2. Left: the comparison of the stress distribution deduced from membrane model (red) and shell model (blue). Right: very different stress distribution among top, middle, and bottom layers shown by shell model (membrane model no longer works); far right shows a diagram of the top, middle, and bottom layers.

Figure R3. (a) Loading curves of the 30L graphene (blue) and 32L BN (red) plus the corresponding fittings (green); (b) good fittings can be obtained only up to small loads.

3. How many times the test was conducted for each layer number? And how about the variation?

Response and changes: The number of the tested sheets (i.e. sample number in statistics) is mentioned as N in the manuscript. For example, “The E^{2D} of 1-3L graphene were $342 \pm 8 \text{ N}\cdot\text{m}^{-1}$ ($N = 11$), $645 \pm 16 \text{ N}\cdot\text{m}^{-1}$ ($N = 13$) and $985 \pm 10 \text{ N}\cdot\text{m}^{-1}$ ($N = 6$), respectively.” The N for 1-3L BN was 11, 14, and 6, respectively. All standard deviations are mentioned in the main text, such as “The E^{2D} of 1-3L graphene were $342 \pm 8 \text{ N}\cdot\text{m}^{-1}$ ($N = 11$), $645 \pm 16 \text{ N}\cdot\text{m}^{-1}$ ($N = 13$) and $985 \pm 10 \text{ N}\cdot\text{m}^{-1}$ ($N = 6$), respectively.” Also, the deviations are shown in Figure 3 and 4, though some of the deviations are too small to be shown clearly. For each sheet, we typically did 5 times of indentation under increasing loads till fracture.

4. Which is the factor that dominates the change of the sliding energy in graphene and BN? in-

plane strain or out-of-plane compression? DFT simulations of pure compression should be made to reveal the mechanisms.

Response and changes: There is an interplay between strain and pressure that affects the sliding energy in graphene and BN. We have done new sets of simulations to show such interplay using a higher-level of theory, i.e. DFT (PBE) plus many-body dispersion (MBD) corrections (PBE+MBD), relative to the one used in the manuscript (opt-B88-vdW functional). Both methods gave similar results.

In the case of graphene, when no strain is applied onto the system, pure pressure (compression) has a key factor in changing the energy profile from barriers near $\sim +7$ meV/unit cell at 0 GPa, to values in the range of ~ -1.5 to -6.0 meV/unit cell at 17.9 GPa (Figure R4a). Several meta-stable energy positions are also observed along the AB-AB path, which indicate strong interactions between p_z -orbitals at different points of the sliding path. We have also performed simulations under pure strain without compression, and the results show a slight increment of the sliding energy barrier (Figure R5a). When both strain and pressure were included (Figure R6a), the sliding energy in graphene further decreased dramatically. These results strongly suggest that both strain and pressure play together in changing the sliding energy in graphene: pressure (compression) changes the sign (from positive to negative) of the energy barrier, while strain increases the magnitude of the negative energy (i.e. further downhill).

In the case of BN, both pure out-of-plane pressure and pure in-plane strain give rise to systematic increases of sliding energy (Figure R4b and R5b), but the combination of pressure and strain

further increases such barrier. In other words, both strain and pressure play together to enhance the sliding energy barriers (Figure R6b). Interestingly, the pressure does not change the sign of the barrier in hBN as it does on graphene. This remarkable different behavior is directly reflected in the different ways that the two materials reacted to indentation.

These new results and discussion have been added to the Supplementary Information, Figure S8-10. In addition, we have added two new sentences in P.14 of the manuscript: “To validate the above results, we also performed simulations at a higher level of theory using DFT (PBE) plus many-body dispersion (MBD) corrections (PBE+MBD).⁷¹ The PBE+MBD results which are shown in Supplementary Information, Figure S8, are fully consistent with those from optB88-vdW functional and from previous works at zero strain and pressure, though numerical differences as high as ~30% in sliding energies between optB88-vdW and PBE+MBD approaches were observed, which evince the accuracy of our simulations. This comparison suggests the generality of the underlying physics associated with the sliding processes, which are not method- or functional-dependent. In addition, we found that there was an interesting interplay between strain and pressure in affecting the sliding energy in graphene and BN (see Supplementary Information, Figure S9 and S10).”

Figure R4: Theoretical calculations on the sliding energy in 2L graphene (a) and BN (b) under pure out-of-plane compression (without in-plane strain).

Figure R5: Theoretical calculations on the sliding energy in 2L graphene (a) and BN (b) under pure in-plane strain (without out-of-plane compression).

Figure R6. Theoretical calculations on the sliding energy in 2L graphene (a) and BN (b) under both in-plane strain and out-of-plane compression.

Reviewer #2 (Remarks to the Author):

This manuscript presents an experimental study of mechanical properties of h-BN layers, both in monolayer and multilayer regime. The key finding is that the mechanical response of h-BN layers is qualitatively different from multilayered graphene, despite the very similar atomistic structure of these two quasi-2D materials. This is attributed to the fact that h-BN layers "stick to each other" much stronger under applied load, while graphene seems to be prone to interlayer sliding, leading to a much less robust mechanical response.

The very different mechanical properties of h-BN and graphene under applied load constitute a very interesting finding that will surely lead to substantial future work. I was certainly

entertained while reading the manuscript and felt that I learnt a few new things. The manuscript is written very clearly and is accessible to a wide audience of chemists and materials scientists. Therefore, I recommend publication in Nature Communications.

I do have a few recommendations regarding the interpretation of DFT calculations that the authors should address before publication:

(1) The experiments seem to be very clear on the different mechanical response of h-BN and graphene. However, the DFT calculations invariably rely on several (many!) approximations, both in the modeling of the actual physical process and in the solution of the electronic structure equations. The authors compare to existing calculations in the literature, and their own calculations agree with some previous results, while they disagree with others. No analysis is attempted to explain this agreement/disagreement. I would have liked to see at least a certain degree of analysis to clarify the situation in the field.

It is often the case that results of "first principles" calculations are taken without any critical analysis. I would expect better from a paper in Nature Communications. The approximations employed in different methods are well known to experts and the authors should strive to make an effort in assessing the reliability of their own and related previous calculations.

Response and changes: We agreed with the reviewer about the limitations that DFT methods normally show in the simulations of several chemical and physical processes. We have initially performed the simulations using a functional (optB88-dW) that inherently does not describe well

the vdW contributions of atoms in their particular chemical environment. This results in numerical differences relative to more accurate methods, e.g. PBE+MBD in Ref,^{65,66} as mentioned by the reviewer in the next comment. The disagreement observed came from the methodologies utilized. We have, however, carried out a completely new set of calculations using PBE+MBD methods as described in details in the next response item.

We have included the following sentences in P14: “To validate the above results, we also did simulations at a higher level of theory using DFT (PBE) plus many-body dispersion (MBD) corrections (PBE+MBD).⁷¹ The PBE+MBD results which are shown in Supplementary Information, Figure S8, are fully consistent with those from optB88-vdW functional and from previous works under the zero strain and pressure condition, though numerical differences as high as ~30% in sliding energies between optB88-vdW and PBE+MBD approaches were observed, which set the top accuracy of our simulations. This suggests that the sliding processes are not method- or functional-dependent, giving generality to the underlying physics involved. In addition, we found that there was an interplay between strain and pressure in affecting the sliding energy in graphene and BN (see Supplementary Information, Figure S9 and S10).

(2) Connected to the previous point, the particular DFT functional that the authors employ (opt-B88-vdW) significantly overestimates interlayer interactions in graphene (and most likely in h-BN). This has been documented in

<http://journals.aps.org/prl/abstract/10.1103/PhysRevLett.115.115501>, for further analysis of the complexities involved in modeling interlayer sliding in graphene and h-BN, see Ref. 69. Either the authors should explicitly mention that their approach is inherently very approximate, or it

would even be better if they added calculations using a less approximate method, such as PBE+MBD to reproduce some of their interlayer sliding potentials. This method is available in the code employed by the authors. If it turns out that the findings remain the same, it would make the paper significantly more interesting for theoretical materials scientists.

Response and changes: We have performed a new set of simulations using the approach suggested by the referee, i.e. PBE+MBD, and the results are shown in Figure R6. The new results agree excellently with the initial calculations using the optB88-vdW functional, though we observed numerical differences between the two methods by a ~30%. It suggests that the observed physics is not method-dependent. This gives generality and breath to the findings and interpretations based on the theoretical methods used in the manuscript. The PBE+MBD results and discussion have been added to the Supplementary Information, Figure S8-10.

(3) The authors comment that no previous study looked at mechanical properties of stacked h-BN and their difference from graphene. However, this is precisely what Ref. 69 did for multilayered h-BN and graphene, albeit in absence of any applied load. Hence, this should be mentioned in the introduction to the paper.

Response and changes: We do acknowledge that the interlayer interactions (without strain or compression) in graphene and BN have been theoretically studied, such as in Ref. 69 (old ref number). However, there are still knowledge gaps: i) P.4 "... the mechanical properties of monolayer BN have never been experimentally examined." The mechanical properties refer to Young's modulus and strength (as mentioned in the review of the previous studies in

Introduction), instead of interlayer interaction. ii) P.12 "...there has been no study on how strain and compression affect their interlayer sliding." Such study has not been reported before, including Ref. 69.

To make it clearer, we added the following discussion in Introduction (P.5): "On the other hand, the different interlayer interactions in few-layer BN and graphene⁶⁵⁻⁶⁷ could play important roles in their mechanical properties." The paper <http://journals.aps.org/prl/abstract/10.1103/PhysRevLett.115.115501> has been added as a new reference (Ref. 67).

Reviewer #3 (Remarks to the Author):

The manuscript describes the differences in mechanical properties and interlayer interaction difference under mechanical load. It shows that although graphene has thickness dependence, h-BN does not. The authors carefully investigated the mechanism behind this phenomenon using experimental and theoretical approaches and concluded that h-BN has stronger interlayer interaction under normal load and tensile strain than graphene. I find that the study is very systematic and clear and highly recommend to be published after a minor revision. I have a few comments for clarification of their arguments.

1. In Figure 1(c), the height profile is drawn in the opposite direction and is confusing to the readers. The profile direction should be reversed.

Response and changes: We thank the Reviewer for pointing this mistake out and have corrected it in the revised manuscript, as shown below:

2. In line 220, they used DFT calculation for sliding energy. They must have applied strain on graphene and h-BN in the calculations. However, there is no mention on the strain directionality. Is it mono-axial or bi-axial strain calculation? Depending on the directionality, the calculation result will be very different.

Response and changes: Bi-axial strain without preferential direction was used in the calculations, consistent with the experimental condition. We have added more description in P.14: “In the vdW-corrected DFT calculations, we chose four combinations of bi-axial strain and hydraulic pressure conditions to reveal the interlayer interactions close to the indentation center of 2L graphene and BN.” and in the Methods: “Bi-axial strain and hydraulic pressure were used in the calculations.”

3. They calculated sliding energy. But there is no mention on the definition of “sliding energy”. They should define it and meaning in this study.

Response and changes: We agree that it is essential to define sliding energy in the manuscript, so we have added one sentence to P.12: “The sliding energy is taken from the total energy differences relative to AB to AB or AA’ to AA’ positions at different points of the sliding pathway.”^{65,66}

Reviewers' Comments:

Reviewer #1 (Remarks to the Author):

The revision with new analyses being added is satisfied. The manuscript can be accepted now.

Reviewer #2 (Remarks to the Author):

The authors have convincingly addressed the comments of all three referees. The manuscript has improved substantially as a result. I recommend publication in Nature Communication as per reasons given in my original report.

Just a quick remark. The author's citations of the MBD method and its implementation in VASP should be updated. In addition to Ref. 73 on which the MBD method is based, they should cite Phys. Rev. Lett. 108, 236402 (2012) [the original MBD method development publication], and

T. Bucko et al., PRB 87, 064110 (2013)

J. Phys.: Condens. Matter 28, 045201 (2016).

These papers report the implementation of the MBD method in the VASP code.

Reviewer #3 (Remarks to the Author):

The authors addressed the comments very well and the manuscript looks improved. I think it is acceptable now as it is.

Dear Dr. Erdonmez,

We thank again the efforts and comments by the three Reviewers. In regard to Reviewer 2's comment:

Reviewer #2 (Remarks to the Author):

The authors have convincingly addressed the comments of all three referees. The manuscript has improved substantially as a result. I recommend publication in Nature Communication as per reasons given in my original report.

Just a quick remark. The author's citations of the MBD method and its implementation in VASP should be updated. In addition to Ref. 73 on which the MBD method is based, they should cite Phys. Rev. Lett. 108, 236402 (2012) [the original MBD method development publication], and T. Bucko et al., PRB 87, 064110 (2013) J. Phys.: Condens. Matter 28, 045201 (2016). These papers report the implementation of the MBD method in the VASP code.

Change: We have added the three papers to the manuscript as Ref. 62-64 (highlighted in yellow in the manuscript with changes highlighted file):

“62. Tkatchenko, A., DiStasio, R. A., Car, R. & Scheffler, M. Accurate and efficient method for many-body van der Waals interactions. Phys. Rev. Lett. 108, 236402 (2012).

63. Bucko, T., Lebegue, S., Hafner, J. & Angyan, J. G. Tkatchenko-Scheffler van der Waals correction method with and without self-consistent screening applied to solids. Phys. Rev. B 87, 064110 (2013).

64. Bucko, T., Lebegue, S., Gould, T. & Angyan, J. G. Many-body dispersion corrections for periodic systems: an efficient reciprocal space implementation. J. Phys. Condens. Matter 28, 045201 (2016).”

Reviewer #1 (Remarks to the Author):

The revision with new analyses being added is satisfied. The manuscript can be accepted now.

Reviewer #3 (Remarks to the Author):

The authors addressed the comments very well and the manuscript looks improved. I think it is acceptable now as it is.

Response: No further action has been taken based on Reviewer 1 and 3's comments.